# Joint Breast Neoplasm Detection and Subtyping using Multi-Resolution Network Trained on Large-Scale H&E Whole Slide Images with Weak Labels

**Adam Casson**[*][1]                                                           ADAM.CASSON@PAIGE.AI
[1] *Paige.AI,*

**Siqi Liu**[*][1]                                                                 SIQI.LIU@PAIGE.AI
**Ran A. Godrich**[1]                                                       RAN.GODRICH@PAIGE.AI
**Hamed Aghdam**[1]                                                     HAMED.AGHDAM@PAIGE.AI
**Donghun Lee**[1]                                                         DONGHUN.LEE@PAIGE.AI
**Kasper Malfroid**[1]                                                   KASPER.MALFROID@PAIGE.AI
**Brandon Rothrock**[1]                                               BRANDON.ROTHROCK@PAIGE.AI
**Christopher Kanan**[2]                                               CKANAN@CS.ROCHESTER.EDU
[2] *University of Rochester*

**Juan Retamero**[1]                                                     JUAN.RETAMERO@PAIGE.AI
**Matt Hanna**[1,3]                                                       MATTHEW.HANNA@PAIGE.AI
[3] *Memorial Sloan Kettering Cancer Center*

**Ewan Millar**[1,4]                                                       EWAN.MILLAR@PAIGE.AI
[4] *University of New South Wales*

**David Klimstra**[1]                                                     DAVID.KLIMSTRA@PAIGE.AI
**Thomas Fuchs**[1]                                                       THOMAS.FUCHS@PAIGE.AI

**Editors:** Accepted for publication at MIDL 2023

## Abstract

Breast cancer is the most commonly diagnosed cancer in the world. The use of artificial intelligence (AI) to help diagnose the disease from digital pathology images has the potential to greatly improve patient outcomes. However, methods for training these models for detecting, segmenting, and subtyping breast neoplasms and other proliferative lesions often rely on costly and time-consuming manual annotation, which can be infeasible for large-scale datasets. In this work, we propose a weakly supervised learning framework to jointly detect, segment, and subtype breast neoplasms. Our approach leverages top-k multiple instance learning to train an initial neoplasm detection backbone network from weakly-labeled whole slide images, which is then used to automatically generate pixel-level pseudo-labels for whole slides. A second network is trained using these pseudo-labels, and slide-level classification is performed by training an aggregator network that fuses the embeddings from both backbone networks. We trained and validated our framework on large-scale datasets with more than 125k whole slide images and demonstrate its effectiveness on tasks including breast neoplasms detection, segmentation, and subtyping.

**Keywords:** Digital Pathology, Multiple Instance Learning, Breast Neoplasm Detection

---

[*] Contributed equally

## 1. Introduction

Breast cancer has surpassed lung cancer as the most commonly diagnosed cancer world-wide, with 2.26 million new cases in 2020 (Ferlay et al., 2021). Early detection and accurate diagnosis is crucial for determining an optimal treatment strategy and improving patient outcomes. Medical institutions have only recently begun the transition from microscopes to digital pathology workflows using high-resolution whole-slide images (WSIs), which introduces tremendous opportunity to incorporate modern AI methods to more thoroughly and systematically help detect cancer, provide segmentation masks to aid in interpretation, and subtyping the detected cancer regions to aid in treatment decision-making (Madabhushi and Lee, 2016; Srinidhi et al., 2021; Tizhoosh and Pantanowitz, 2018). Training AI algorithms for digital pathology is challenging due to the large size of WSIs and limited availability of ground truth data. To accommodate large image dimensions and keep GPU memory usage tractable, digital pathology AI systems generally break WSIs into small tiles and aggregate tile information for slide-level predictions (Campanella et al., 2019; Lu et al., 2021; Li et al., 2021). The ground truth data is often only available as a summary diagnostic at the case or specimen level, making it labor-intensive and expensive to create pixel or object level ground truth at an industrial scale. Most studies in the field have only focused on small-scale datasets such as (Litjens et al., 2018; Zuley et al., 2016), which may not reflect the complexity and variability of real-world distributions, making it difficult to determine the effectiveness of algorithmic differences when large-scale datasets are available. To address these challenges, a framework is needed to leverage weak ground truth extracted from clinical diagnostic reports to scale training and evaluation to these large-scale datasets.

**Contribution:** In this paper, we propose a joint breast cancer detection, segmentation, and subtyping framework that only relies on clinical diagnostic reports for training, and does not require additional manual annotation. Our approach decomposes the system training into 3 stages. In stage 1, it uses top-k multiple instance learning (Campanella et al., 2019) to train an initial neoplasm detection backbone network, which is then used to generate pseudo-labels for regions of interest. In stage 2, a second network is trained to subtype the image regions at a coarse granularity using pseudo-labels. In stage 3, slide-level classifications are obtained by a per-slide aggregator network that fuses the embeddings from the stage 1 and 2 networks. Our method is trained and validated using 125k+ H&E WSIs. Segmentation performance is evaluated using 105 WSIs exhaustively labeled with different cancer subtypes by expert pathologists. Additionally, we compute the RoI-level classification accuracy on the BRACS dataset (Brancati et al., 2021) without finetuning. We further evaluate the specimen level neoplasm detection performance on 7k+ slides and investigate different options of obtaining the WSI-level prediction results.

## 2. Related work

There have been many pioneering studies that have explored the use of deep learning systems on breast hematoxylin and eosin (H&E) WSIs (Chan et al., 2023). For example, in Cruz-Roa et al. (2017), a convolutional neural network (CNN) was employed for detecting invasive tumors on WSIs. Similarly, Gandomkar et al. (2018); Mi et al. (2021) used a deep network to classify WSIs as benign or cancerous, and then further categorized cancer cases into subtypes. Le et al. (2020) employed deep CNNs to quantitatively assess tumor and tumor-

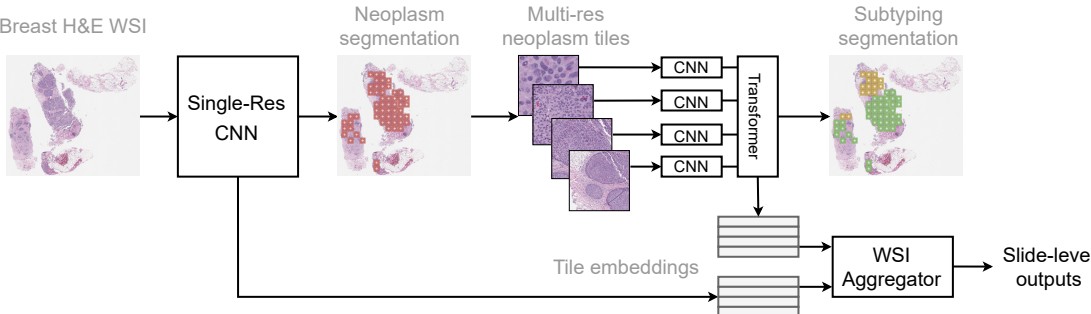

Figure 1: The overview of the proposed framework.

infiltrating lymphocytes, and Wang et al. (2022) explored the use of deep learning models to directly assist in the histological grading of breast cancer. These studies demonstrate the potential of utilizing deep learning techniques to improve the diagnostic accuracy and efficiency of breast cancer detection and analysis. Pati et al. (2022) explored using a hybrid graph network to classify the breast neoplasm into fine-grained subtypes and evaluated on the BACH (Aresta et al., 2019) and BRACS datasets (Brancati et al., 2021). A few studies have employed pseudo-labeling techniques, similar to those used in our study to map diagnosis reports to pixel-level ground truth for prostate cancer grading (Bulten et al., 2020; Marini et al., 2021; Silva-Rodriguez et al., 2021). These methods generally assigned the pure prostate grading labels, e.g., 3+3, to all the regions with detected cancer. Besides the analogy between prostate cancer grading and breast neoplasm sub-typing, it is challenging to confirm pure breast slides with only one subtype of breast neoplasm based on the clinical reports since less severe findings are often intentionally omitted by the pathologist. We adopted the pseudo-labeling to breast neoplasm subtyping by (1) automatically separating the invasive cancer regions with the rest (2) merging the in-situ and atypia neoplasms to mitigate the impact of the label noise. In Marini et al. (2022), authors attempted to generate weak WSI-labels from free-text clinical reports. Recent studies (Li et al., 2021; van Rijthoven et al., 2021; Sandbank et al., 2022; Hashimoto et al., 2020; Karimi et al., 2020) have begun to investigate the use of multi-resolution inputs for convolutional neural networks (CNNs) in digital pathology analysis. This approach enables the CNN to capture both fine-grained details at high resolution and contextual information at lower resolutions. Additionally, in Chen et al. (2022), the authors expanded the receptive field of tile networks with a hierarchical vision transformer trained with self-supervised learning.

## 3. Methods

Our proposed framework consists of three successive networks shown in Figure 1. The first CNN is trained using a top-k MIL framework (Campanella et al., 2019) on specimen level labels to detect neoplasm tiles in slides. Although MIL algorithms perform well on slide-level subtyping, it is challenging to obtain multi-class mutually exclusive segmentation. This is because MIL algorithms are trained explicitly to make slide-level predictions whereas obtaining multi-class mutually exclusive segmentation requires tile-level multi-class

predictions. In the second stage of training, we therefore train a network using the tile-level ground truth generated by the pseudo-labelling technique. For pseudo-labelling, the top-k MIL trained CNN is used for selecting training tiles for subtyping, which are assigned to the same slide-level labels. In practice, we observe it sometimes requires a much larger spatial context than that needed for cancer detection to distinguish certain types of breast neoplasms, e.g., high-grade DCIS vs. invasive cancer, and we therefore train a multi-resolution CNN on these tiles using the pseudo-labels. To fuse information from both CNNs to produce the final slide-level predictions, we use an attention-based aggregator network that take the embeddings from both CNNs as inputs. The ground truth used in this study was automatically parsed from the clinical diagnosis reports, containing 6 common breast neoplasms at a specimen level: Invasive breast carcinoma of no special type (previously known as invasive ductal carcinoma or IDC), Invasive lobular carcinoma (ILC), Ductal carcinoma in situ (DCIS), Lobular carcinoma in situ (LCIS), Atypical ductal hyperplasia (ADH), and Atypical lobular hyperplasia (ALH) (Lakhani et al., 2012). The specimen is labeled as benign if none of the 6 subtypes or other rare cancer types were reported.

### 3.1. Breast Neoplasm Detection Trained with Otsu Top-$k$ MIL

We first train a SE-ResNet-50 (Hu et al., 2019) using a top-k multiple instance learning (MIL) strategy based on Campanella et al. (2019) to detect non-benign tiles in a slide. The foreground regions of the WSI are extracted using an HSV filter with fixed HSV ranges tuned for non-fat breast tissues in H&E WSIs. At the start of each training epoch, the CNN first scores each foreground tile extracted at 0.5 $\mu$m / pixel (MPP), 20X in general, with the cancer subtype presence probabilities with sigmoid outputs. For each training step, the CNN weights are only updated using the top $k$ highest ranked instances for each neoplasm subtype. Though the tile-wise ground truth labels for the non-benign slides can be noisy, the top $k$ tiles picked by CNN on benign slides are reliable. The optimization is thus mainly driven by the benign slides and was shown to be effective when training with large-scale datasets (Campanella et al., 2019). To adapt to lesions of varying sizes, instead of using a fixed $k$ value in Campanella et al. (2019), we used the Otsu's method (Otsu, 1979) on the tile predictions to dynamically choose the $k$ for each slide while setting a maximum cap $k_{max} = 400$ fixed in our experiments. After training, an approximated segmentation map can be generated for each subtype by thresholding the network sigmoidal outputs.

### 3.2. Breast Neoplasm Subtyping trained with Pseudo-Labeling

The top-k MIL CNN described above does not enforce mutually exclusive scores for breast neoplasm subtypes. To address this, we train a second CNN specialized in subtyping the neoplasm tiles detected by the detection CNN. We use pseudo-labelling to generate ground truth labels for slide tiles. For invasive cancer specimens, we assume the tiles covered by the invasive cancer segmentation, obtained from the top-k MIL network, all belong to the same invasive cancer subtype. (2) for non-invasive cancer specimens which were diagnosed to have either atypia or in-situ neoplasm, we assume all the regions underneath the breast neoplasm segmentation belong to either ADH / DCIS or ALH / LCIS. In clinical reports, pathologists will often omit reporting atypical findings when a more severe diagnosis is present. Thus, we merged the in situ and atypia subtypes and formulated breast subtyping as a 4-class

classification problem involving IDC, ILC, DCIS/ADH, or LCIS/ALH. For this training phase, we exclude benign slides and slides having both ductal and lobular neoplasms. The subtyping CNN is then trained to predict the mutually exclusive probabilities that a tile center resides in one of four neoplasm subtypes.

It is important to consider larger contexts in breast neoplasm subtyping, since pathologists often toggle between magnifications to check the lesions boundaries and other macroscale morphological patterns for breast cancer diagnosis. To address this issue, we use a multi-resolution CNN network, as illustrated in Figure 1. Each training instance contains multiple tiles of different sizes sharing the same tile center, which are then rescaled to the same size. Each tile is processed separately by a SE-ResNet-50. In our experiments, the multi-scale CNNs do not have shared weights. The resulting embeddings are treated as tokens and fused by a small transformer network with fixed positional encoding (PE) (Vaswani et al., 2017). The network output is a softmax function, assuming that the tile center class labels are mutually exclusive. 2D coarse-grained subtype labelmaps can be derived from the network outputs using $argmax$ of the tile-level subtype probabilities. Though the multi-res CNN takes more compute than single-res CNN to process each individual tile, only the detected neoplasm tiles are fed into it during both training and inference.

### 3.3. Attention based aggregator for slide-level prediction

Given that the detection and subtyping backbone networks can each provide feature embeddings for WSI tiles, we use an aggregator network to fuse the embeddings and obtain the final slide-level predictions. Since the presence of each neoplasm is not mutually exclusive at slide-level by nature, the network outputs a binary prediction for each class. We explored two approaches: (1) training an aggregator network using only the tile embedding from the detection network; (2) or training an aggregator network to fuse the tile embeddings from both the detection and subtyping networks. For the latter, we only extract subtyping embeddings on tiles detected with breast neoplasm to avoid tiles that are out-of-distribution for the subtyping network. When there is no neoplasm detected, a zero vector is used as a placeholder. We use all tissue tiles for extracting detection embeddings. Section 5.3 details the network architecture choices. The slide-level outputs are activated with a sigmoid function. To map the slide-level predictions to the specimen level ground truth, we take the maxpool of the sigmoid outputs across all the slides in the same specimen.

### 4. Data

We used a dataset of over 125k WSIs with mixed biopsies and excision specimens, which were acquired from the Memorial Sloan Kettering Cancer Center as described in Table 1. The majority of the slides were scanned using Leica AT2 and GT450 scanners. Sequential cases were deidentified and randomly sampled to represent a natural patient population. The train and validation splits were created at specimen level with random sampling therefore sharing a similar distribution. The specimen ground truth labels were obtained by automatically parsing the diagnostic reports. The validation set was used for hyperparameter tuning and model selection. We reserved 2000 specimens for the test set used for performance reporting. We randomly sampled 188 slides from the non-benign test slides that were exhaustively annotated by board certified pathologists with labelled polygons

Table 1: The summary of the in-house dataset used in this study showing number of slides and specimens. We also break down the number of specimens for neoplasm subtypes and benign according to the clinical diagnosis.

| Split | Slides | Specimens | IDC | ILC | DCIS | LCIS | ADH | ALH | Benign |
|---|---|---|---|---|---|---|---|---|---|
| Train | 93369 | 44683 | 5209 | 905 | 9724 | 2968 | 3245 | 2812 | 28262 |
| Validation | 24020 | 7235 | 813 | 154 | 1542 | 432 | 508 | 429 | 4655 |
| Test | 7989 | 2000 | 272 | 51 | 453 | 151 | 147 | 151 | 1222 |
| Total | 125378 | 53918 | 6294 | 1110 | 11719 | 3551 | 3900 | 3392 | 34139 |

to analyze segmentation performance. We reserved 83 slides as a tuning set for operating point selection and 105 slides for evaluation. For external evaluation of RoI based breast subtyping without finetuning, we used the entire BRACS dataset (Brancati et al., 2021).

## 5. Results

### 5.1. Breast Neoplasm Localization

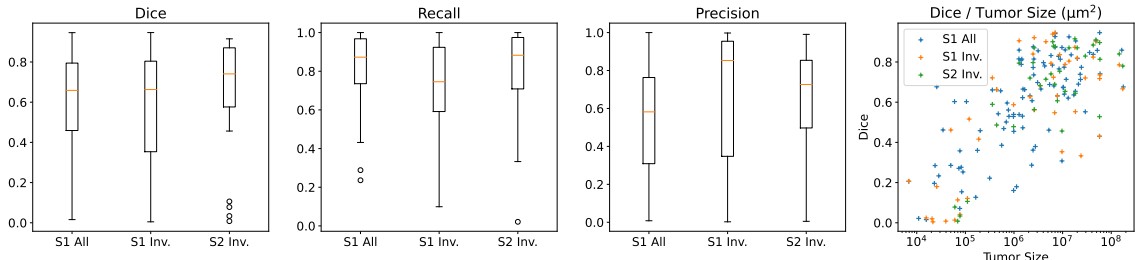

Figure 2: The plots of pixel-wise dice, recall and precision of the cancer segmentation including all breast neoplasms (S1 All) or only the invasive cancer areas (S1 Inv. and S2 Inv.). All the dice scores are plotted with the tumor size on the right.

To evaluate the localization of all breast neoplasm detection and invasive breast cancer, we obtained the all neoplasm and invasive segmentation masks from the backbone networks as described in Section 3.1 and Section 3.2. The invasive breast cancer segmentation can be obtained from either the detection or subtyping network. In Figure 2, we show the box plots of per-slide dice, recall and precision of 105 test WSIs. In summary, the invasive cancer segmentation (S2 Inv.) from the sub-typing network has better average dice ($0.66 \pm 0.24$) than the invasive cancer segmentation (S1 Inv.) from the detection backbone ($0.56 \pm 0.30$). It also surpassed the dice scores ($0.56 \pm 0.30$) of the all neoplasm segmentations (S1 All) from the detection network, although the areas of invasive cancer are larger. As also noted in Reinke et al. (2022), we also show that larger tumors tend to have higher dice scores in general on the right of Figure 2.

Table 2: Ablative analysis for the components of the breast neoplasm subtyping network.

| Architecture | MPP=0.5 | MPP=2.0 | MPP=4.0 | MPP=10.0 | Positional Enc. | Tile-Wise Accuracy |
|---|---|---|---|---|---|---|
| SE-ResNet-50×1 | ✓ | ✗ | ✗ | ✗ | ✓ | 0.74 |
| SE-ResNet-50×2 | ✓ | ✓ | ✗ | ✗ | ✓ | 0.78 |
| SE-ResNet-50×3 | ✓ | ✓ | ✓ | ✗ | ✓ | 0.77 |
| SE-ResNet-50×4 | ✓ | ✓ | ✓ | ✓ | ✗ | 0.79 |
| SE-ResNet-50×4 | ✓ | ✓ | ✓ | ✓ | ✓ | **0.81** |

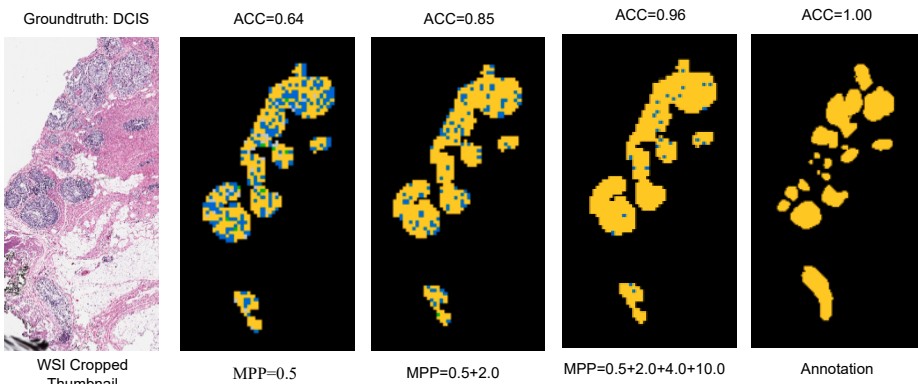

Figure 3: Visual examples showing a WSI crop where adding more contexts progressively improved the tile-wise subtyping tile-wise accuracy (ACC). Different colors in the segmentation labelmaps indicate different breast neoplasm subtype categories.

## 5.2. Breast Neoplasm Subtyping

An ablative analysis was performed on 105 annotated slides with mixed breast neoplasm subtypes to investigate the contributions of different components of the proposed neoplasm sub-typing network as shown in Table 2. Since the neoplasm region areas vary within and between WSIs, we computed the tile-wise classification accuracy (ACC) within each slide and averaged it among the whole dataset as the performance indicator. By adding larger contexts, we observed an increase of ACC from 0.74 to 0.81 except adding $MPP = 4.0$ on top of higher resolutions did not show improvement. By removing the position encoding, which provides the ordering of resolutions to the resolution fusion transformer, we noticed the accuracy dropped from 0.81 to 0.79. As shown in Table 3, we also benchmarked the subtyping network on both the test splits and the entire BRACS dataset (Brancati et al., 2021) without finetuning. The required context size $10 \times 10 \mu m^2$ is mostly larger than the

Table 3: The F1 scores computed on the the BRACS dataset RoIs. Please note that the fine-grained subtyping results from Pati et al. (2022) are not directly comparable to ours.

| Split | Network | Training method | Normal | Benign | Atypia | In-Situ | Invasive |
|---|---|---|---|---|---|---|---|
| Test | CNN 0.5 mpp (Pati et al., 2022) | Fully Supervised | 0.420 | 0.423 | 0.227 | 0.503 | 0.770 |
| Test | MS-CNN 1.0+0.5+0.25 mpp (Pati et al., 2022) | Fully Supervised | 0.503 | 0.443 | 0.317 | 0.573 | 0.860 |
| Test | HACT-Net (Pati et al., 2022) | Fully Supervised | 0.615 | 0.474 | 0.404 | 0.664 | 0.884 |
| All | MS-CNN 10.0+4.0+2.0+0.5 mpp | Weakly supervised | 0.876 | | 0.685 | | 0.731 |
| Test | MS-CNN 10.0+4.0+2.0+0.5 mpp | Weakly supervised | 0.902 | | 0.528 | | 0.826 |

annotated BRACS RoIs. To predict the RoI classification labels, we located each RoI in the corresponding WSI and extracted the context window needed by our network at the center of the previous RoI. If the RoI is not within the cancer detection segmentation, it is then classified as Normal / Benign. We quoted the F1 scores from Table 5 of Pati et al. (2022) as reference. Though we had a slightly different label grouping setup compared to Pati et al. (2022), the F1 scores of Invasive cancers can be directly compared. On the test set, though our proposed network was trained with weakly supervised pseudo-labelling without being finetuned on the BRACS training data, it reached an F1 score of 0.826, which is higher than fully supervised single-scale CNN (0.77) and lower than a fully supervised multi-scale CNN (0.86). In Pati et al. (2022), the hybrid nuclei graph network based HACT-Net achieved a higher F1 score (0.884) than the image CNN based methods. Within the same 3-stage framework, it would be interesting to explore the feasibility of training similar graph-network based architectures using the proposed large-scaled pseudo-labeling in our future work. Though not directly comparable, we provide fine-grained subtyping numbers from Pati et al. (2022) for reference. The proposed framework achieved a high F1 score (0.902) for the Normal / Benign because our framework decomposes the tile classification problem into detection and subtyping separately.

### 5.3. Breast Neoplasm Detection

Specimen level AUC scores for detecting each breast neoplasm subtype are given in Table 4. As a baseline, we take the maximum tile scores from the detection network to represent the specimen level prediction (Maxpool MIL). With the tile embedding from the detection network only, we studied the attention-based MIL aggregators similar to Attention MIL (Ilse et al., 2018) and DSMIL (Li et al., 2021). As shown in Figure 5, we also evaluated aggregator architectures to fuse the tile embedding from both the detection and subtyping networks. In each architecture, the outputs of the attention branches are concatenated and then fed into the output layer. For DSMIL architectures we experimented with the source of the critical instances used as the query vectors in the attention layers. The critical instances are determined by selecting the tiles with the highest probability for each class. We tried three approaches: (1) using the detection embeddings to determine critical instances where the selected embeddings would then be used as the query vectors for attention on both the detection and subtyping embeddings; (2) using the subtyping embeddings in the same manner; (3) using both detection and subtyping embeddings to select two sets of critical instances to be used independently in their own attention layers. We notice an improvement in AUC compared to the Maxpool MIL baseline for all aggregators, except in benign AUC, with the largest improvement seen in ADH (0.896 vs 0.951). The performance on invasive carcinomas is similar between the two overall approaches ($\geq$0.983 AUC for all learned aggregators), however we found that inclusion of the subtyping embeddings can improve AUC on the non-invasive lesions, specifically ADH (0.951), DCIS (0.982), and LCIS (0.965).

## 6. Conclusion

Here, we proposed a framework for jointly detecting, segmenting, and subtyping breast neoplasms, which requires no manual annotation for training. We trained and validated the framework using over 125k WSIs. Our subtyping network improved the segmentation of

Table 4: The specimen level detection AUCs for each breast neoplasm subtype.

| Name | ADH | ALH | DCIS | LCIS | IDC | ILC | Inv. | Benign | Inv. vs. Other |
|---|---|---|---|---|---|---|---|---|---|
| Positive | 147 | 151 | 453 | 151 | 272 | 51 | 387 | 1206 | 387 |
| Total | 1617 | 1553 | 1950 | 1695 | 1931 | 1931 | 2000 | 2000 | 794 |
| **Detection Embedding** | | | | | | | | | |
| Maxpool MIL | 0.896 | 0.946 | 0.951 | 0.932 | 0.976 | 0.971 | 0.976 | **0.986** | 0.945 |
| Attention MIL | 0.950 | **0.969** | 0.980 | 0.962 | **0.986** | **0.994** | **0.984** | 0.984 | 0.962 |
| DSMIL (Li et al., 2021) | 0.943 | 0.965 | 0.978 | 0.963 | 0.985 | 0.989 | 0.983 | 0.982 | 0.958 |
| CLAM (Lu et al., 2021) | 0.950 | 0.968 | 0.977 | 0.959 | 0.982 | 0.994 | 0.981 | 0.984 | 0.952 |
| **Detection Embedding + Subtyping Embedding** | | | | | | | | | |
| Attention MIL | **0.951** | 0.966 | **0.982** | 0.961 | 0.984 | 0.992 | 0.983 | 0.984 | 0.955 |
| DSMIL (cross attention 1) | 0.938 | 0.962 | 0.978 | **0.965** | **0.986** | **0.994** | **0.984** | 0.979 | 0.961 |
| DSMIL (cross attention 2) | 0.941 | 0.959 | 0.973 | 0.952 | 0.985 | 0.993 | **0.984** | 0.981 | **0.963** |
| DSMIL (separate attention) | 0.950 | 0.965 | **0.982** | 0.959 | 0.984 | **0.994** | 0.983 | 0.983 | 0.956 |

invasive cancer regions and using a multi-resolution network improved tile-level cancer classification accuracy. Without training with any manual pixel-level annotations, our weakly supervised subtyping network achieved F1 scores on-par with fully supervised CNNs trained with slides collected from unseen data sources. We evaluated slide-level detection AUCs using different aggregator architectures, observing an improvement over using the detection network alone.

**Acknowledgements**   We would like to thank our talented team of pathologists who contributed to this study by providing their medical expertise and feedback to validate the application scenarios and the overall study design. We are especially grateful to Joe Oakley from Paige AI, NY US and Jorge S. Reis-Filho from Memorial Sloan Kettering Cancer Center, NY US. We would also like to acknowledge the contributions of Patricia Raciti and Felipe Geyer to this project during their time at Paige.

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

## Appendix A. Network architectures

### A.1. Detection network

For the SE-ResNet-50 (Hu et al., 2019) detetion network used in Section 3.1, we change the last convolutional layer to output 512 channels instead of the original 2048 with a projection

layer. The final SE block was also modified to retain the constant reduction ratio of $r = 16$ throughout the network. Lastly, we replaced the final global averaging pooling layer with max pooling.

## A.2. Subtyping network

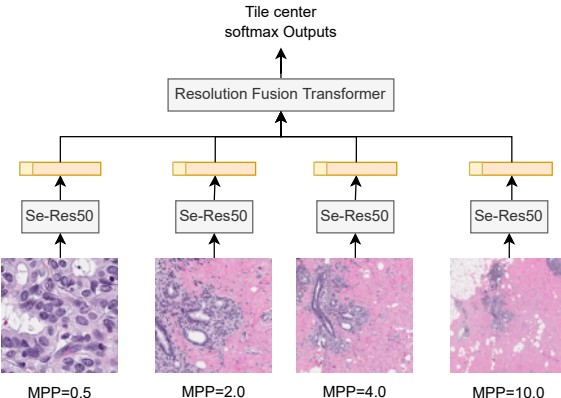

Figure 4: Illustration of the multi-res backbone architecture.

We use four separate SE-ResNet-50 networks, one for each magnification level, to extract features of size 2048. These networks follow the original SE-ResNet-50 implementation without the modifications used in the detection network. We use a standard Transformer network (Vaswani et al., 2017) with 3 encoder layers. We create the classifier input by flattening the output of the Transformer to obtain a single embedding of size 8192.

## A.3. DSMIL variants

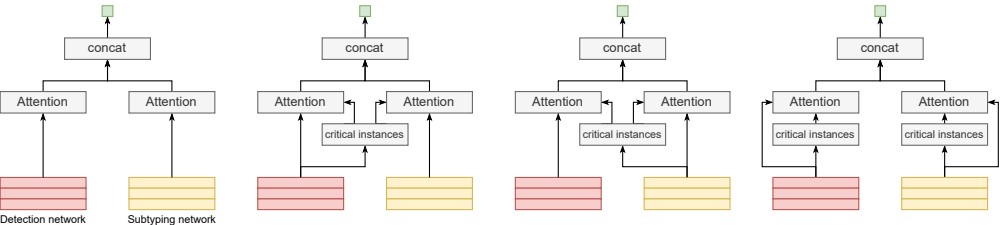

Figure 5: Architectures for fusing detection and subtyping embeddings. From left to right: 1) attention MIL, 2) DSMIL with detection embeddings as critical instances, 3) DSMIL with subtyping embeddings as critical instances, and 4) DSMIL with separate critical instances.

The DSMIL architecture used only the detection embedding follows the architecture described in (Li et al., 2021). The modified architecture variants that use both detection and subtyping embedding are presented below.

Let $\mathbf{x}_d \in \mathbb{R}^{T \times 512}$ be our 512-dimensional detection embedding for all $T$ tissue tiles, and let $\mathbf{x}_s \in \mathbb{R}^{N \times 8192}$ be the 8192-dimensional subtyping embeddings for all $N$ tiles predicted to

be neoplastic by the detection network. For the first variant, we use the detection embedding for critical instance selection. The detection embedding are fed through a linear layer with 7 output units corresponding to the classes ADH, ALH, DCIS, LCIS, IDC, ILC, and all invasive, i.e.,

$$\mathbf{i} = \sigma(\mathbf{W}_i \mathbf{x}_d), \tag{1}$$

where $\sigma$ is the logistic sigmoid function. The output of this branch is connected to a binary cross entropy loss for optimizing tile-level predictions. Max-pooling the tile-level probabilities leads to selecting the embedding with the highest probability for each class which are considered the critical instances $\mathbf{c}$ such that $\mathbf{c} \subseteq \mathbf{x}$. The critical instances are then used as queries in scaled dot product attention with the full set of detection embedding and subtyping embedding in separate attention heads:

$$
\begin{aligned}
\mathbf{q}_d &= \tanh(\mathbf{W}_a(\text{relu}(\mathbf{W}_e \mathbf{c})) & \mathbf{q}_s &= \tanh(\mathbf{W}_a(\text{relu}(\mathbf{W}_c \mathbf{c})) \\
\mathbf{k}_d &= \tanh(\mathbf{W}_a \text{relu}(\mathbf{W}_e \mathbf{x}_d)) & \mathbf{k}_s &= \tanh(\mathbf{W}_a \text{relu}(\mathbf{W}_e \mathbf{x}_s)) \\
\mathbf{v}_d &= \mathbf{W}_v \mathbf{x}_d & \mathbf{v}_s &= \mathbf{W}_v \mathbf{x}_s \\
\mathbf{z}_d &= \text{softmax}(\frac{\mathbf{q}_d \mathbf{k}_d^T}{\sqrt{d_{k_d}}})\mathbf{v}_d & \mathbf{z}_s &= \text{softmax}(\frac{\mathbf{q}_s \mathbf{k}_s^T}{\sqrt{d_{k_s}}})\mathbf{v}_s
\end{aligned}
$$

where despite the shared notation for the weight matrices, the weights are not shared between the attention heads. For all models we set $\mathbf{W}_a$ to have 64 output units and all other weight matrices are square except for the output layers. The slide-level class predictions are obtained by concatenating the output of the attention followed by a linear layer and a sigmoid function:

$$\mathbf{b} = \sigma(\mathbf{W}_b[\mathbf{z}_d; \mathbf{z}_s]). \tag{2}$$

For the second variant, where we use the subtyping embedding for critical instances, the roles of $x_d$ and $x_s$ would simply be swapped in the notation above. The third version of the network uses both detection and subtyping embedding to compute two sets of critical instances which requires calculating two set of tile-level predictions:

$$
\begin{aligned}
\mathbf{i}_d &= \sigma(\mathbf{W}_{id} \mathbf{x}_d) \\
\mathbf{i}_s &= \sigma(\mathbf{W}_{is} \mathbf{x}_s)
\end{aligned}
\tag{3}
$$

where we max-pool with respect to each class to select the critical instances in each source of embedding $\mathbf{c}_d$ and $\mathbf{c}_s$ such that $\mathbf{c}_d \subseteq \mathbf{x}_d$ and $\mathbf{c}_s \subseteq \mathbf{x}_s$. The critical instances are the queries for independent attention heads along with their respective source embedding. The output of the network follows Equation (2). We also apply dropout ($p = 0.25$) to the value vectors in the attention heads and on the concatenated attention head outputs as well.

## Appendix B. Optimization details

### B.1. Detection network

The model was trained for 20 epochs. The training dataset consisted of 938 million unique tiles of size $224 \times 224$ taken from the $20\times$ magnification level (0.5 $\mu$m/px) of the WSIs. We

applied random image augmentations to the tiles including rotations, vertical/horizontal flips, color-jittering, sharpening, blurring, and contrast stretching. We used Adam optimizer (Kingma and Ba, 2017), learning rate of 4e-5, and effective batch size of 1536.

### B.2. Subtyping network

We applied random image augmentations to the tiles including vertical/horizontal flips, color-jittering, sharpening, and contrast stretching. We used Adam optimizer (Kingma and Ba, 2017), learning rate of 1e-5, and effective batch size of 128.

### B.3. Aggregators

The labels for our dataset were parsed from clinical diagnosis reports which summarize findings at the specimen level. Due to this, along with memory constraints caused by the large-scale nature of breast WSIs, we used a two-pass training algorithm. We first make a forward pass without gradients, over all slides in a given specimen. We then take the slide with the highest probability as the representative slide for the specimen and compute a second forward pass with gradients computed. All aggregator models were trained using AdamW optimizer (Loshchilov and Hutter, 2019), learning rate of 5e-4, and per-device batch size of 1 with gradient accumulation tuned according the number of devices to achieve a consistent effective batch size of 32. We used early stopping monitoring the validation loss.

## Appendix C. Example Breast Neoplasm Segmentation Visualizations

In accompaniment to Figure 2, which presents the localization numbers for breast neoplasm and invasive cancer segmentation, we provide a visual demonstration of the framework's performance through examples. Figure 6 depict cases where the framework achieved a Dice score higher than 0.8, indicating successful capture of the entire annotated region with minimal false positives. However, as seen in Figure 7 and Figure 8, the framework can also exhibit typical error modes, such as missing small cancer regions or over-segmentation of the biopsy site changes, which often leads to false positives.

To further illustrate the capabilities of the framework, we present two examples of invasive cancer segmentation in Figure 9 and Figure 10. These examples demonstrate the difference between the segmentation obtained from the detection network (S1 Inv.) and that obtained from the subtyping network (S2 Inv.). Through this comparison, it becomes evident that the subtyping network is able to improve both the sensitivity and specificity of the invasive cancer segmentation, as observed in our results. To reflect average results, we also show randomly selected segmentation examples in Figure 11 for breast neoplasm segmentation and Figure 12 for invasive cancer segmentation.

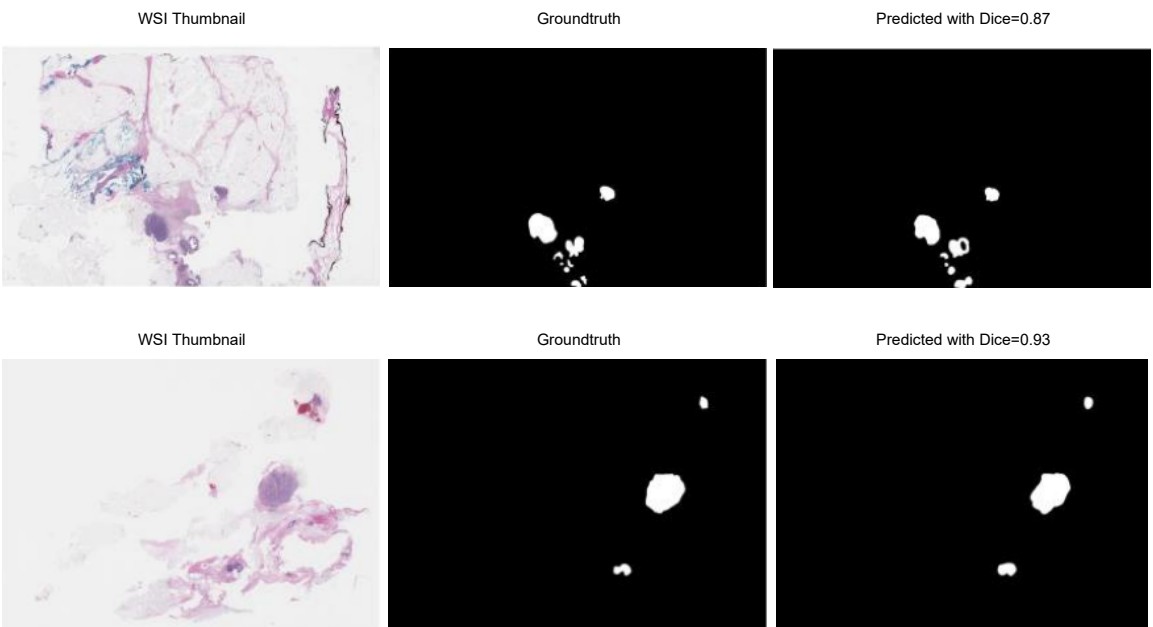

Figure 6: Example breast neoplasm segmentation with high dice scores.

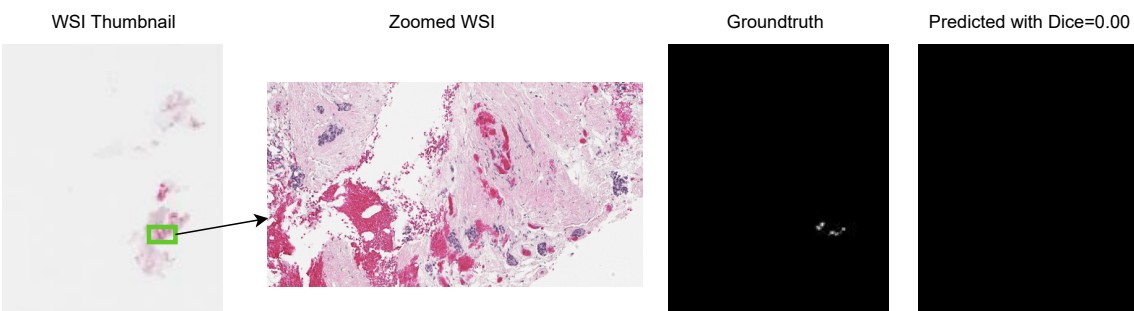

Figure 7: The only occurrence of a breast neoplasm segmentation with 0 dice score in our results, which likely occurred due to the small size of the cancer region.

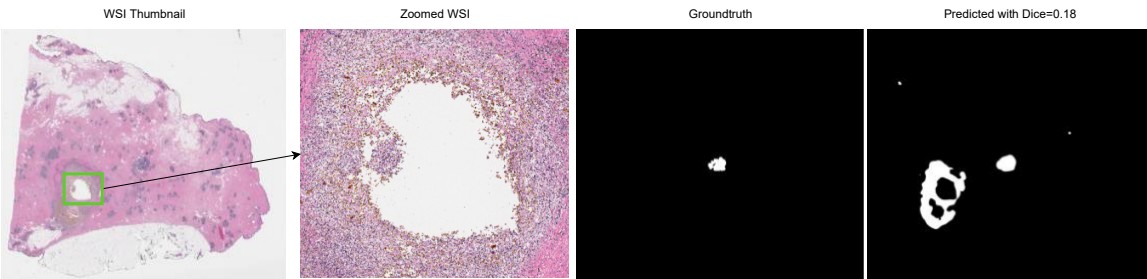

Figure 8: Example breast neoplasm segmentation with 0.18 dice score. The network over-segmented with biopsy site.

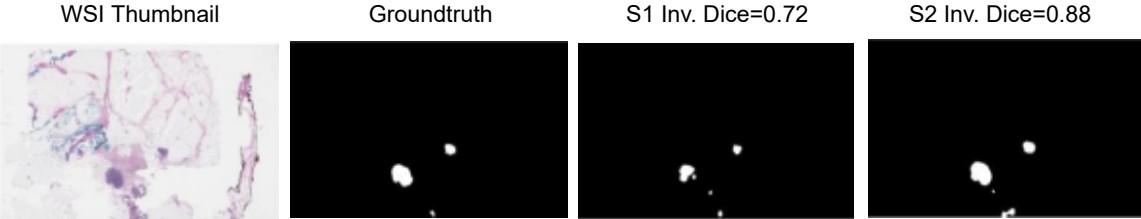

Figure 9: Example of invasive cancer segmentation where the invasive cancer segmentation generated from the subtyping network (S2 Inv.) has higher sensitivity and dice score compared to the one obtained from the detection network (S1 Inv.).

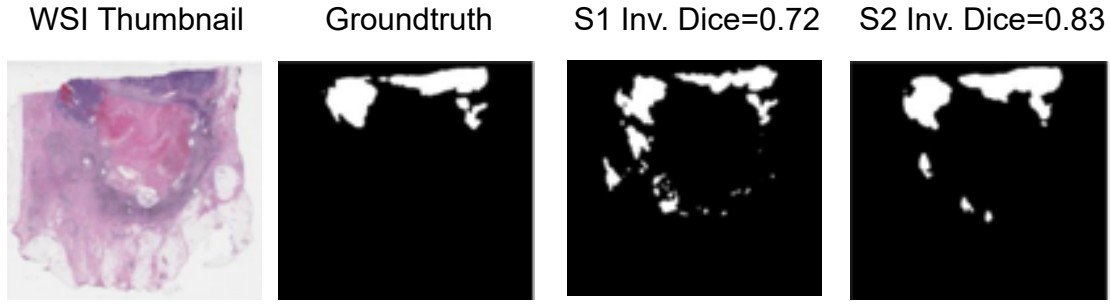

Figure 10: Example of invasive cancer segmentation where the invasive cancer segmentation generated from the subtyping network has less false positives.

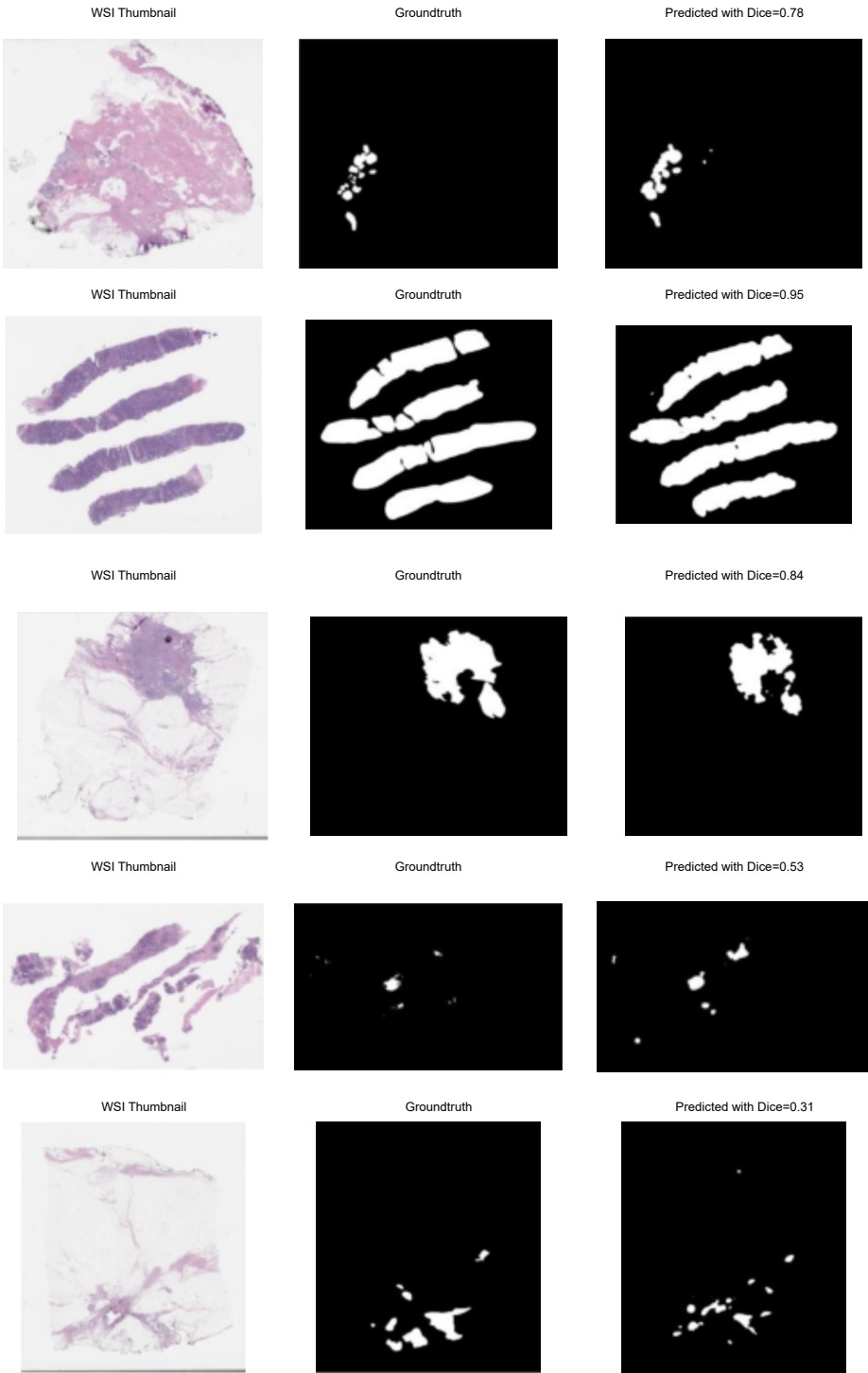

Figure 11: Randomly sampled examples of breast neoplasm segmentation.

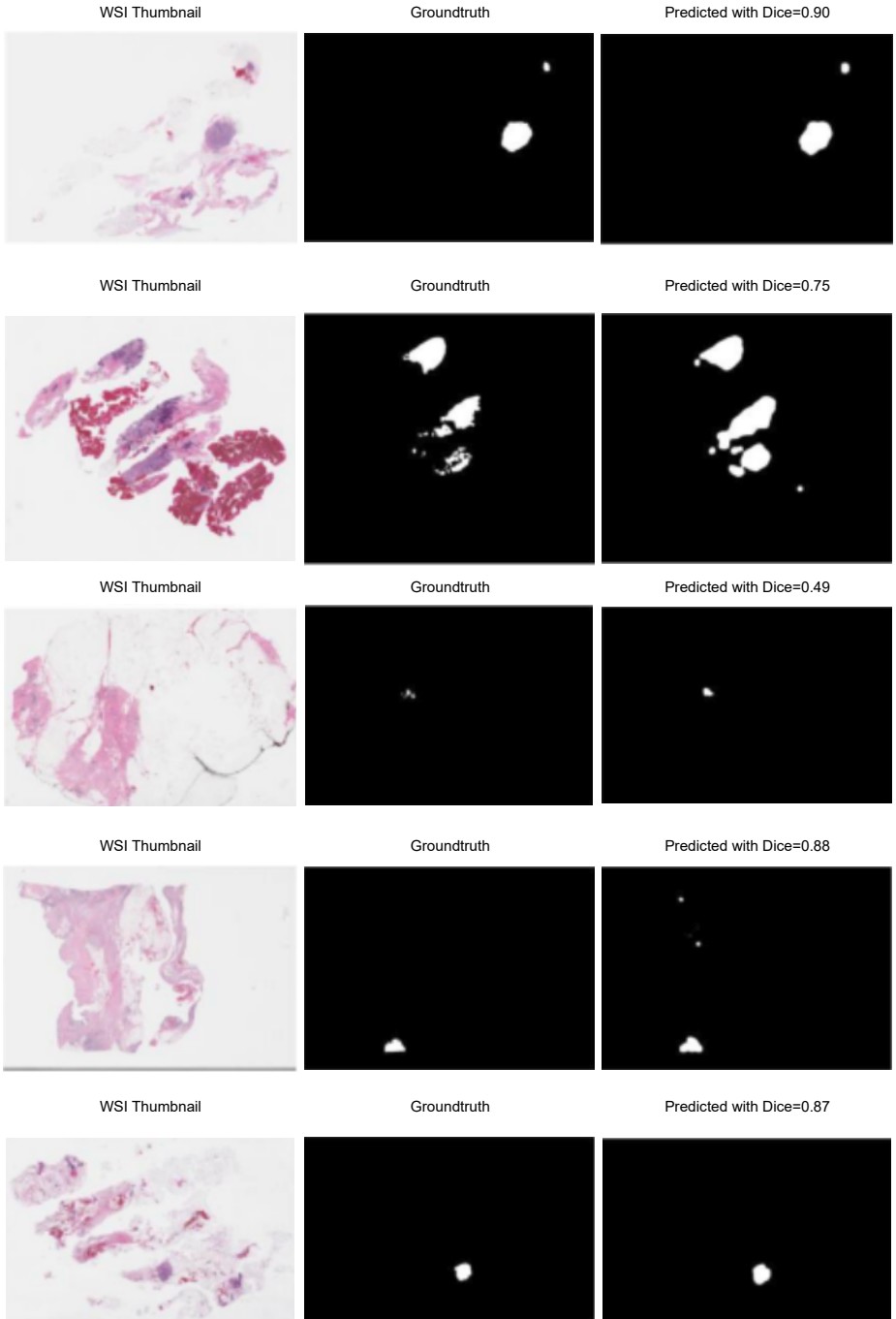

Figure 12: Randomly sampled examples of breast neoplasm segmentation.

