# OpenReview forum: "Joint Breast Neoplasm Detection and Subtyping using Multi-Resolution Network Trained on Large-Scale H&E Whole Slide Images with Weak Labels"
_MIDL.io/2023/Conference — MIDL 2023 Oral_

### Official Review · Reviewer_2Scw · 2023-02-01

**Confidence:** 4
**Preliminary Rating:** 5
**Recommendation:** Oral

**Summary:**

The authors present an approach for training a breast lesion segmentation and classification using slide-level annotations. Their approach comprises 3 stages: First, they use a MIL approach to generate tile-level pseudo labels from the slide-level annotations. Secondly, they train a subtyping network using the generated pseudo labels. Lastly, they aggregate the embeddings extracted from the first two stages to predict slide-level classes.

**Strengths:**

* The paper addresses the important issue that for large histology datasets usually only slide-level annotations are available
* The paper is well written and easy to follow
* Data set is large and also the evaluation data set seems to be of appropriate size to support their findings
* The methods used are appropriate

**Weaknesses:**

* In Section 5.2, the authors compare the F1 scores for the invasive class. They claim that their results (0.826) is comparable to the baseline (0.770 to 0.860). However, the third baseline has a higher score, which is not mentioned. Also, the baseline scores vary quite a lot and it is hard to compare a single value to 3 spread out values. This should be addressed in the text
* Figures could be more clear to help the reader understand the whole analysis pipeline. Also, Figures 1 and 2 are hard to read because of the small size
* Introduction needs some revision
	* The contributions are already partly stated in the paragraph before the contribution paragraph and then repeated
	* The beginning of the introduction lacks references
	* "Recent systems break WSIs in to small tiles [...] Campanella et al., 2019". Firstly, it is not only recent but most approaches do that. Secondly, at least 2 papers should be referenced to proof the plural "recent systems". Thirdly, 2019 is not so recent.

**Deanonymize Review:**

no

**Detailed Comments:**

minor points:
* The first sentence in the abstract is too long and complex. Readers might stop reading right there.
* Introduction "an weakly"
* The size of the data set is referred to as 100k+ or 125k+, should be consistent throughout the paper
* The abbreviation "PE" is not explained in Table 2
* Table 3 has F1 scores > 1 for all baseline atypia and in-situ values. Surely percentage is meant, but should be fixed

**Paper Type:**

methodological development

**Questions To Address In The Rebuttal:**

The authors should address the weaknesses and minor points given above. The point regarding the F1 scores for the invasive class are most important in my view. The minor points are more or less quick fixes.

---

### Official Review · Reviewer_MMEZ · 2023-02-02

**Confidence:** 4
**Preliminary Rating:** 4
**Recommendation:** Poster

**Summary:**

The authors have proposed a weakly supervised learning framework to detect, segment, and subtype breast neoplasm. They have used a multi-stage approach to classify disease subtypes accurately. First, they used a top-k MIL approach to detect neoplasm patches in the slides. Next, the top-k probable neoplasm patches from training data are combined with the slide’s subtype labels and used for training a multi-resolution detection model. Finally, the learned embeddings of the first and second stages are fused via an attention-based aggregation approach for slide-level prediction. The authors validated the approach on large-scale in-house breast cancer data containing more than 125K WSIs and six common breast neoplasms. Further, they reported their performance on the BRACS dataset for comparison on open-sourced data. The models achieved a high F1-score for subtype classification and demonstrated a strong ability to localize the tumor using weak labels.

**Strengths:**

- The paper targets a relevant disease type - breast cancer and aptly motivate the problem.
- Authors have incorporated multiple novel ideas to stitch together their final frameworks, such as the top-k MIL framework, attention pooling, and multi-resolution approach. Further, they have empirically validated the inclusion of each modeling architecture.
- Authors have used large-scale data containing more than 125K WSIs and 6 disease subtypes for validation. Further, medical experts have annotated 188 WSIs at a granular level for evaluating the segmentation approach.
- In table 4, the authors have provided a detailed comparison of different pooling approaches.
- The weakly-supervised segmentation approach is giving similar F1 score to the fully-supervised approach.


**Weaknesses:**

- The authors have not explained the top-k MIL approach in detail, making it difficult to understand the incorporated parts of Campanella et al.. Further, different top-k MIL approaches have been proposed since Campanella et al.. Hence it is necessary to clarify the author’s used top-k MIL approach.
- Many better-performing WSI weakly-supervised learning approaches, such as CLAM, have been proposed since Campanella et al.. In the paper, it is unclear why the authors used the top-K MIL approach instead of a recent superior-performing framework.
- The method section is not well-structured and motivated. Authors have used multiple novel approaches but haven’t shared their rationale for including each component. For instance, an attention-based MIL approach performs robustly for cancer subtypes. Then why did the authors decide to propose a new multi-stage approach?
- The paper proposes a multi-stage approach, and an ablation study for different components has been provided. However, there is no proper comparison to publically available SOTA approaches, such as CLAM, TransMIL, etc.
- There are floating point issues in table-3.


**Deanonymize Review:**

no

**Detailed Comments:**

The motivation for the problem is well written in the paper. However, the motivation for using the author's proposed method is unclear. It will be helpful if the authors can highlight the gap in the existing literature and specify the contribution of their paper. Also, kindly let me know if I have missed any relevant detail in the paper. And it might be helpful also to include those details in the paper.

**Paper Type:**

methodological development

**Questions To Address In The Rebuttal:**

Please address the points highlighted in the weakness. The paper needs to clarify the gap in the existing literature that authors are filling by proposing their new approach. Further, performance comparison to other recent approaches will be helpful in showcasing the impact. The method doesn't necessarily need to be the best among all, but it should be competitive and well-motivated. Also, it might be helpful if authors could combine Figures 1 and 2 to include a stage-wise flow of their approach.

---

### Official Review · Reviewer_zpW1 · 2023-02-03

**Confidence:** 4
**Preliminary Rating:** 4
**Recommendation:** Oral

**Summary:**

The authors propose a weakly supervised method for segmentation and subtype classification of breast neoplasms in H&E images.
The method can be decomposed into three blocks:
1) Network 1: neoplasm detection from weakly-labeled whole slide images (labels from clinical diagnostic reports). Trained with top-k multiple instance learning.
-> generate pixel-level pseudo-labels for whole slides with only one subtype.
2) Network 2: Multi-resolution model trained using these pseudo-labels to subtype the image regions at a coarse granularity.
3) Slide-level classification is performed by training a per-slide aggregator network that fuses the embeddings from both networks.

The paper is well written. The model is trained and validated (with manual annotations) on >100K H&E WSIs. RoI- level classification is also validated on BRACS dataset.

**Strengths:**

- The evaluation is done on a very large (private) dataset.
- The method is interesting and seems sound.
- Relatively good results are obtained, with ablation study, and comparison with SoA on the external BRACS dataset


**Weaknesses:**

Nothing major. Some similar works are mentioned in the related work (Bulten et al., 2020; Marini et al., 2021; Silva-Rodriguez et al., 2021). It should be explained what is missing from these methods and what is novel in the proposed approach to overcome it.

Other minor comments/weaknesses below.


**Deanonymize Review:**

no

**Detailed Comments:**

“larger tumors tend to have higher dice scores in general” see
Reinke, Annika, Lena Maier-Hein, and Henning Müller. Common limitations of performance metrics in biomedical image analysis. No. CONFERENCE. 2021.

Report standard deviations when possible.

Table 3: Atypia and In-situ results for the related works should be 0.227, 0.503 etc. instead of 22.67 etc.

Table 5: Briefly mention/explain in the caption why the fine-grained subtyping results are not comparable with the related works.

Maybe missing literature
Marini, Niccolò, et al. "Unleashing the potential of digital pathology data by training computer-aided diagnosis models without human annotations." NPJ digital medicine 5.1 (2022): 102.

Lu, Ming Y., et al. "Data-efficient and weakly supervised computational pathology on whole-slide images." Nature biomedical engineering 5.6 (2021): 555-570.

Hashimoto, Noriaki, et al. "Multi-scale domain-adversarial multiple-instance CNN for cancer subtype classification with unannotated histopathological images." Proceedings of the IEEE/CVF conference on computer vision and pattern recognition. 2020.

**Paper Type:**

methodological development

**Questions To Address In The Rebuttal:**

I think it is a strong method and contribution and I am happy with the content of the paper. Some of the minor points/weaknesses I mentioned above should be addressed to improve the quality of the paper.

---

### Official Review · Reviewer_v75C · 2023-02-03

**Confidence:** 5
**Preliminary Rating:** 4
**Recommendation:** Poster

**Summary:**

This article presents a weakly supervised learning approach for simultaneous detection, segmentation, and classification of breast tumors. The model training involves three stages, where the outputs from the earlier stages provide supervision for the later ones. Additionally, multi-resolution tiles are utilized to incorporate global context into decision making. The study also investigates the impact of different components on the model performance.

**Strengths:**

1. The paper is easy to follow.
2. Through the model training involves three phases, the combination of these phases appears to complement each other and achieves plausible results in terms of detection and segmentation.
3. The utilization of multi-resolution processing seems to enhance the overall performance.

**Weaknesses:**

1. Regarding the second phase of Breast Neoplasm Subtyping, four separate Se-ResNet50 models are trained to extract features from tiles of different resolutions, which incurs a significant computational burden. Can you elaborate on the training process? It is not specified how the training sets for each model are generated. Is it necessary to use different models if the goal is to obtain multi-resolution embeddings, and why weight sharing is not feasible?
2.  Still in the second phase of Breast Neoplasm Subtyping, the authors assume that all neoplasm tiles belong to the same subtype. Is this assumption valid? If not, what impact would it have on the final results? Further discussion on this point would be appreciated.
3. There is a lack of visual results for Breast Neoplasm Localization. Although some are included in the supplementary material, they present a favorable scenario and may not reflect average results. Can you provide an illustration of typical outcomes?
4. The experimentation section lacks comparison with previous works, making it difficult to place this study within the broader context of the field.

**Deanonymize Review:**

no

**Detailed Comments:**

Please refer to the comments in the Weakness section.

**Paper Type:**

methodological development

**Questions To Address In The Rebuttal:**

1. Regarding the second phase of Breast Neoplasm Subtyping, four separate Se-ResNet50 models are trained to extract features from tiles of different resolutions, which incurs a significant computational burden. Can you elaborate on the training process? It is not specified how the training sets for each model are generated. Is it necessary to use different models if the goal is to obtain multi-resolution embeddings, and why weight sharing is not feasible?
2.  Still in the second phase of Breast Neoplasm Subtyping, the authors assume that all neoplasm tiles belong to the same subtype. Is this assumption valid? If not, what impact would it have on the final results? Further discussion on this point would be appreciated.
3. There is a lack of visual results for Breast Neoplasm Localization. Although some are included in the supplementary material, they present a favorable scenario and may not reflect average results. Can you provide an illustration of typical outcomes?
4. The experimentation section lacks comparison with previous works, making it difficult to place this study within the broader context of the field.

---

### Meta-Review · Area_Chair_vFTb · 2023-02-23

**Recommendation:** Accept (Oral)
**Confidence:** 5

**Metareview:**

This paper proposed a weakly supervised method for segmentation and subtype classification of breast neoplasms in H&E images. The method is interesting and experimental results on large datasets demonstrated the efficacy of the methods. Minor concerns have been raised by the reviewers. Based on the consensus from all reviewers, a decision of accept is recommended.